# Metagenomic and Metabolomic Analyses Reveal the Role of Gut Microbiome-Associated Metabolites in the Muscle Elasticity of the Large Yellow Croaker (*Larimichthys crocea*)

**DOI:** 10.3390/ani14182690

**Published:** 2024-09-16

**Authors:** Zhenheng Cheng, Hao Huang, Guangde Qiao, Yabing Wang, Xiaoshan Wang, Yanfeng Yue, Quanxin Gao, Shiming Peng

**Affiliations:** 1College of Life Sciences, Huzhou University, 759 Erhuan East Road, Wuxing District, Huzhou 313000, China; chengzh0628@163.com (Z.C.); huanghao2638@126.com (H.H.); 2East China Sea Fishery Research Institute, Shanghai 200090, China; qiaogd@ecsf.ac.cn (G.Q.); wangyabing@ecsf.ac.cn (Y.W.); wangxs@ecsf.ac.cn (X.W.); yueyf@ecsf.ac.cn (Y.Y.)

**Keywords:** *Larimichthys crocea*, muscle elasticity, gut microbiota, metabolome, metagenome

## Abstract

**Simple Summary:**

We identified a large number of core gut flora and key metabolites associated with muscle elasticity by integrating macrogenomic and metabolomic techniques, thus providing insights into the molecular mechanisms explaining the differences in muscle quality in the large yellow croaker (LYC, *Larimichthys crocea*). We further constructed an association model between gut flora and metabolites, which offers a new perspective on the potential mechanisms by which the LYC gut flora influences muscle quality via the regulation of the intestinal metabolism. Additionally, our study not only provides new insights into the gut microbiota–metabolism–muscle axis, but also strong support for future research on muscle quality trait improvement in LYC.

**Abstract:**

The large yellow croaker (LYC, *Larimichthys crocea*) is highly regarded for its delicious taste and unique flavor. The gut microbiota has the ability to affect the host muscle performance and elasticity by regulating nutrient metabolism. The purpose of this study is to establish the relationship between muscle quality and intestinal flora in order to provide reference for the improvement of the muscle elasticity of LYC. In this study, the intestinal contents of high muscle elasticity males (IEHM), females (IEHF), and low muscle elasticity males (IELM) and females (IELF) were collected and subjected to metagenomic and metabolomic analyses. Metagenomic sequencing results showed that the intestinal flora structures of LYCs with different muscle elasticities were significantly different. The abundance of Streptophyta in the IELM (24.63%) and IELF (29.68%) groups was significantly higher than that in the IEHM and IEHF groups. The abundance of *Vibrio scophthalmi* (66.66%) in the IEHF group was the highest. Based on metabolomic analysis by liquid chromatograph-mass spectrometry, 107 differentially abundant metabolites were identified between the IEHM and IELM groups, and 100 differentially abundant metabolites were identified between the IEHF and IELF groups. Based on these metabolites, a large number of enriched metabolic pathways related to muscle elasticity were identified. Significant differences in the intestinal metabolism between groups with different muscle elasticities were identified. Moreover, the model of the relationship between the intestinal flora and metabolites was constructed, and the molecular mechanism of intestinal flora regulation of the nutrient metabolism was further revealed. The results help to understand the molecular mechanism of different muscle elasticities of LYC and provide an important reference for the study of the mechanism of the effects of LYC intestinal symbiotic bacteria on muscle development, and the development and application of probiotics in LYC.

## 1. Introduction

The large yellow croaker (LYC, *Larimichthys crocea*), one of the most economically important marine fish in China, has received widespread attention worldwide because of its tasty meat and nutritious characteristics, thus becoming one of the four major marine economic fishes in China [1,2]. The LYC is specifically distributed in the coastal and neighboring waters of China, and the primary culture mode is net-pen culture [2,3]. In recent years, the culture area and production of LYC has grown continuously, and it has become an essential part of China’s mariculture industry [4,5]. Nonetheless, the continuously expanding scale of LYC aquaculture has resulted in problems including germplasm degradation, diseases, and quality decline, which have seriously restricted the development of the LYC industry [6,7]. With the increase of living standards and the change in consumer preferences, larger numbers of consumers have begun to pursue high-quality LYC products. Thus, increasing the muscle quality of LYC has become a popular research topic [8].

Weight gain in fish is primarily attributed to the accumulation of lipids and proteins in their muscles [9]. In fish, muscles are the largest tissue and constitute the major part used for human food [10]. Textural properties (hardness, elasticity, chewability, cohesion, and adhesion) are conventional indicators that are widely used to evaluate fish quality [11,12]. Muscle elasticity directly affects the texture of fish products [13]. It has been demonstrated that muscle elasticity in LYC is affected by various factors, including feed, genetics, environment, and culture management [14,15].

Research has demonstrated that gut microbes are closely associated with host growth and development [16], immunomodulation [17,18], nutritional metabolism [19,20], and muscle quality [21,22]. The gut flora influences muscle tissue synthesis by regulating the host’s digestive, absorptive, and total metabolism [23]. Recent research has demonstrated that the presence of the gut flora reduces muscle water loss [24], thereby increasing muscle elasticity and quality. The gut flora and its metabolites also affect the musculoskeletal system function via gut–brain–muscle interactions, ultimately affecting muscle health and quality. Additionally, the gut flora, bile acids, and their receptors interact to form the “gut flora–bile acid” axis [25], which regulates many physiological processes, such as carbohydrate, lipid, and energy metabolism, thus influencing muscle quality [26,27].

To date, there have been few studies on the regulation of muscle elasticity by the intestinal flora in LYCs, and the role played by the interaction between the intestinal flora and nutrient metabolism in the muscle elasticity in LYC has not been reported [28]. Therefore, there is a need for an in-depth exploration of the mechanism of intestinal flora regulation of muscle elasticity. In this research, non-targeted metabolomic and macrogenomic techniques were used to analyze the global structure of the intestinal flora and nutrient metabolism, aiming to provide a detailed analysis of the molecular mechanisms underlying the differences in muscle elasticity in LYCs. We can speculate that the gut microbiota of LYCs could regulate nutrient metabolism to affect muscle quality. Therefore, our results help to explain the molecular mechanism of the differences in muscle elasticity in LYCs and will provide a significant reference for further research into the mechanism of action of LYC intestinal commensal bacteria and the development and application of probiotics.

## 2. Materials and Methods

### 2.1. Sample Collection

The 72 LYCs (Fujian Fuding, China) used in the study were seven months old and were from the same batch acquired from the Fujian Fuding Research Center, East China Sea Research Institute, Chinese Academy of Fisheries Sciences. All farmed LYCs obtained from the same nursery were cultured in the same sea area, using the same feed (Fujian Fuding, China). The geographic coordinates of the sea area: 120.40431, 27.222208. The LYCs were anesthetized using 20 mg/L eugenol solution (Shanghai, China), weighed, and 4 cm^3^ (2 cm × 2 cm × 1 cm) of muscle tissue about 1 cm below the middle of the dorsal fin was collected, frozen in liquid nitrogen, and then stored in the refrigerator at −80 °C for subsequent experimental analyses. Determination of texture characteristics (muscle elasticity): Take the back muscle of large yellow croaker and cut the back muscle into 2 cm × 2 cm × 1 cm squares. The elasticity of the muscle was measured with a TMS-PRO texture analyzer (Beijing Yingsheng Hengtai Technology Co., Ltd., Beijing, China). Extrusion test parameters are as follows: probe pause time 1 s, range of force sensing element 1000 N, probe rebound height 25 mm, shape variable 50%, measuring speed 60 mm/min, starting force 1.5 N. Based on the muscle elasticity data, the samples were classified into four groups: high muscle elasticity males (IEHM), low muscle elasticity males (IELM), high muscle elasticity females (IEHF), and low muscle elasticity females (IEHF) (IEHM: 1.73 mm ± 0.07 mm, IELM: 0.76 mm ± 0.06 mm, IEHF: 1.61 mm ± 0.09 mm, and IELF: 0.81 mm ± 0.10 mm) (Figure 1), and six LYCs from each group were randomly selected to collect their intestinal contents, which were subsequently sent to Shanghai Paysono Bio-technology Co., Ltd. (Shanghai, China) for microbial macrogenomic analysis and non-targeted metabolomic analysis.

### 2.2. Sex Identification Method

The fin samples from LYC were collected in 2 mL centrifuge tubes and preserved in anhydrous ethanol. A DN37-Marine Animal Tissue Genomic DNA Rapid Extraction Kit (Aidlab Biotechnologies Co., Ltd., Beijing, China) was used to extract the DNA, and 1% agarose gel electrophoresis was used for quality testing [29]. Sex identification was performed by PCR amplification based on the single nucleotide polymorphism (SNP) loci reported by Lin et al. (2018), using the following the primer sequences: 6F (5′-ATCTGTCAACCACTGTATCATCTG-3′), 6R (5′-GGATGGCGTTTGGCTGAG-3′) and 6T(5′-CATCCCCAGACCTCCACT-3′) [30]. The amplification products were identified using 1.5% agarose gel electrophoresis (specific conditions: voltage 220 volts, time 25 min, buffer type 1× TAE). The specific conditions of PCR amplification were as followed: Predenaturation 95 °C, denaturation 94 °C, annealing temperature 64 °C, extension 72 °C, cycle 29 times, re-extension 72 °C, preservation 4 °C.

### 2.3. Metagenomic Analysis

Microbial macrogenome analysis was performed by Shanghai Parsonage Biotechnology Limited (Shanghai, China). Total microbial DNA was extracted using a Mag-Bind Soil DNA Kit from Omega Biotek (Winooski, VT, USA). The concentration of DNA was determined using a Qubit 4 Fluorometer system (Qubit™ 4 Fluorometer, with WiFi: Q33238; Qubit™ Assay Tubes: Q32856; Qubit™ 1X dsDNA HS Assay Kit: Q33231, all from Invitrogen, Waltham, MA, USA), and the quality of the DNA was examined using 1% agarose gel electrophoresis [31]. The concentration of the DNA was adjusted using DNA storage solution and stored at −20 °C. The desired genomic library was constructed using the standard Illumina TruSeq DNA Sample Preparation Guide, followed by DNA sequencing on the Illumina platform (Illumina, San Diego, CA, USA). The data were analyzed using principal coordinate analysis (PCoA), non-metric multidimensional scaling (NMDS) analysis, and principal component analysis (PCA) [32].

### 2.4. Metabolome Analysis

Metabolomics analyses were performed by Shanghai Parsonage Biotechnology Co., Ltd. (Shanghai, China) using an untargeted metabolomics approach [33]. After the intestinal samples were thawed slowly at 4 °C, about 40 mg of intestinal sample was taken and added to methanol and acetonitrile, followed by vortex mixing, and left to stand for 10 min [34]. The samples were then centrifuged at 14,000× *g* for 20 min at 4 °C, and the supernatant was collected and dried under a vacuum. For the mass spectrometry analysis, 100 mL of aqueous acetonitrile (acetonitrile: Water = 1:1, vv) was added for reconstitution, vortexed, and centrifuged again at 14,000× *g* for 15 min at 4 °C. The supernatant was collected for metabolite analysis. An Agilent 1290 Infinity LC ultra-high performance liquid chromatography (UHPLC) HILIC column was used in this experiment (Agilent, Santa Clara, CA, USA) [35]. Quality control (QC) samples were inserted into the sample queue to monitor and evaluate the stability of the system and ensure the reliability of the experimental data. The raw data were pre-processed using XCMS software (https://xcmsonline.scripps.edu/landing_page.php?pgcontent=mainPage, (accessed on 17 July 2024)) [36]. Metabolites were identified by accuracy mass and mass spectrometry analysis and matched with standard databases. Orthogonal Projections to Latent Structures Discriminant Analysis (OPLS-DA) and PCA were performed on the data. Biologically significant differentially abundant metabolites (DAMs) were identified based on the criteria variable importance of projection (VIP) > 1 and *p* < 0.05 Finally, the DAMs screened in positive and negative ion modes were combined for pathway analysis.

### 2.5. Statistical Analysis

IBM SPSS Statistics 27 software (IBM Corp., Armonk, NY, USA) was used to determine the mean, standard deviation, and for coefficient of variation analysis of the muscle elasticity data. The data are expressed as means ± standard deviations (SD) [37].

LYC, large yellow croaker; IEHM, intestinal contents of high muscle elasticity males; IELM, intestinal contents of low muscle elasticity males; IEHF, intestinal contents of high muscle elasticity females; IELF, intestinal contents of low muscle elasticity females.

## 3. Results

### 3.1. Composition and Structure of the Intestinal Flora of LYCs

It is impossible to distinguish male and female LYCs by morphology; therefore, we used SNP detection to identify the sex of the LYCs (Figure 2). The structural composition of male and female LYC gut flora at the phylum level is illustrated in Figure 3a,b. In all samples, the dominant phylum was Proteobacteria, whose abundance was higher in the high muscle elasticity group, accounting for 78.50% and 84.53% of the intestinal flora in IEHM and IEHF groups, respectively. Compared with the IELF group, the IEHF group had higher abundance of Proteobacteria, Actinobacteriota, and Streptophyta.

At the genus level, the dominant intestinal bacteria in the male LYCs were Vibrio, which accounted for 74.53% and 46.66% of the total intestinal bacteria in the IEHM and IELM groups, respectively; the abundance of Vibrio in the IEHM group was significantly higher than that in the IELM group. Trifolium and Oryza in the IELM group accounted for 7.08% and 7.57% of the total intestinal bacteria, respectively; however, these two bacterial genera were not detected in the IEHM group (Figure 3a). Additionally, the dominant bacterium in the IEHF group was Vibrio, which accounted for 83.01% of the intestinal flora, and the Vibrio level in the IEHF group was significantly higher than that in the IELF group (15.38%). The mean abundances of Oryza (1.37%) and Trifolium (1.46%) in the IEHF group were significantly lower than those in the IELF group (9.77%, 8.64%) (Figure 3b).

At the species level, *Vibrio scophthalmi* was the dominant intestinal bacterium in the high muscle elasticity groups, accounting for 60.30% and 66.66% of the total intestinal bacteria in IEHM and IEHF groups, respectively, which were significantly higher than those in the IELM group (36.11%) and the IELF group (9.78%) (Figure 3c,d). Overall, there was a significant difference in the gut flora structure between the high and low muscle elasticity groups.

As shown in Figure 3g,h, the results of diversity index analysis showed that the diversity of the intestinal flora in the high and low muscle elasticity groups differed significantly. The Chao1, Simpson, Shannon, and observed_species indices in IEHM group were higher than those in IELM group, which indicated that the diversity of intestinal flora in high muscle elasticity group was higher than that in the low elasticity group for male LYCs (Figure 3e). The Simpson and Pielou_e indices were higher in the IEHF group than in the IELF group, while the Chao1 and observed_species indices were lower in the IEHF group than in the IELF group, which indicated that the difference in diversity of the intestinal flora between the high and low elasticity groups of female LYC muscles was not significant (Figure 3f). In addition, Goods coverage index was greater than 96.5%, indicating that the sequences obtained in this study represent the majority of gut bacteria in each sample.

### 3.2. Analysis of Differences in the Compositional Structure of Intestinal Flora in LYCs with Different Muscle Elasticities

Venn diagrams at the genus level for the gut flora demonstrated that the number of IEHM, IELM, IEHF and IELF groups were 21, 53, 42 and 67, with 79 genera shared by the IEHM and IELM groups, and 81 genera shared by the IEHF and IELF groups (Figure 4a,e). The differences in the gut flora composition of LYCs with different muscle elasticities were visualized using NMDS, PCoA, and PCA. These analyses showed that there was a significant separation between the IEHM and IELM groups (Figure 4b–d), indicating that there are significant differences in the gut flora structure between the high and low muscle elasticity groups in males. The PCoA and PCA analyses showed the significant separation between the IEHF and IELF groups, while the NMDS analysis demonstrated non-significant separation (Figure 4f–h).

To further reveal the differences in the intestinal microbial communities among the groups, the top 10 phyla, genera, and species in terms of average abundance were selected to plot a heat map. The results showed that at the phylum level, the Firmicutes, Chordata, and Thermoplasmatota were more abundant in the IEHM group, while the Basidiomycota, Ascomycota, Proteobacteria, Streptophyta, Firmicutes, Actinobacteriota, and Bacteroidota were more abundant in the IEHF group (Figure 5a). Compared with that in the IELF group, Proteobacteria were more abundant in the IEHF group, while the abundance of Chordata, Thermoplasmatota, Actinobacteriota, Streptophyta, Firmicutes, Bacteroidota, Ascomycota, and Basidiomycota was higher in the IELF group (Figure 5b). Overall, the abundance of Firmicutes was higher in the intestines of both male and female LYCs with high muscle elasticity, while the Basidiomycota abundance was higher in the intestines of both male and female LYCs with low muscle elasticity.

At the genus level, the abundance of Danio was higher in the IEHM group, whereas the abundances of Vibrio, Trifolium, Bradyrhizobium, Azonexus, Oryza, Robertmurraya, Zea, Solirubrobacter, and Streptomyces were higher in the IELM group (Figure 5c). The abundance of Vibrio was higher in the IEHF group, while the abundances of Danio, Zea, Robertmurraya, Trifolium, Azonexus, Solirubrobacter, Bradyrhizobium, Oryza, and Streptomyces were higher in the IELF group (Figure 5d). The abundance of Vibrio was higher in the intestines of male and female LYCs with high muscle elasticity, whereas the abundance of Oryza was higher in the intestines of male and female LYCs with low muscle elasticity.

At the species level, *Vibrio ichthyoenteri*, *Vibrio scophthalmi*, *Solirubrobacter sp003344625*, *Streptomyces clavifer*, *Trifolium repens*, *Oryza sativa*, *Robertmurraya kyonggiensis*, *Zea mays*, *Bradyrhizobium sp003020115*, and *Azonexus sp009469615* showed higher abundance in the IEHM group, whereas in the IELM group, *Zea mays*, *Vibrio scophthalmi*, *Arthrobacter D sp002929375*, and *Microbacterium arborescens* showed higher abundance (Figure 5e). The abundances of *Trifolium repens*, *Zea mays*, *Robertmurraya kyonggiensis*, *Azonexus sp009469615*, *Solirubrobacter sp003344625*, *Bradyrhizobium sp003020115*, *Oryza sativa*, and *Streptomyces clavifer* were higher in the IEHF group, whereas the abundances of *Danio rerio*, *Rhizoctonia solani*, and *Microbacterium arborescens* were higher in abundance in the IELF group (Figure 5f). The difference in the gut flora structures between the high and low muscle elasticity groups was significant.

### 3.3. Metabolite Identification of LYC Intestinal Contents

To further reveal the intestinal metabolic mechanisms underlying the differences in muscle elasticity in LYCs, LC-MS/MS untargeted metabolic analysis of the intestinal contents was performed. Based on the mzCloud, mzVault and MassList databases, a total of 2482 metabolites was identified, including: 756 organic acids and derivatives; 792 lipids and lipid-like molecules; 308 organoheterocyclic compounds; 205 benzenoids; 174 organic oxygen compounds; 94 phenylpropanoids and polyketides; 48 organic nitrogen compounds; 72 nucleosides, nucleotides, and analogs; 20 alkaloids and derivatives; 8 lignans, neolignans, and related compounds; 4 organosulfur compounds; and 1 organohalogen compound.

According to the PCA results (Figure 6a,b), the first principal component (PC1) could explain 44.8% and 21.5% of the features of the original dataset in the IEHM and IELM groups, respectively; and the second principal component (PC2) could explain 26.2% and 42.2% of the features of the original dataset in these two groups. PC1 could explain 36.6% and 35.3% of the features of the original dataset in the IEHF and IELF groups, respectively (Figure 6c,d), and PC2 could explain 25.3% and 27.4% of the features of the original data set in the two groups (Figure 6c,d). Overall, there was a clear separation of principal components between the high and low muscle elasticity groups, indicating significant differences in gut metabolites between the groups.

### 3.4. Analysis of Differential Metabolites of LYC Intestinal Contents

A total of 207 DAMs with significant changes (VIP > 1 and *p* < 0.05) were identified in the male and female LYCs with high and low muscle elasticities. Between the IEHM and IELM groups, a total of 93 DAMs were identified (30 upregulated and 63 downregulated, Figure 6e), and 114 DAMs were identified between the IEHF and IELF groups (51 upregulated and 63 downregulated, Figure 6f). Taken together, the female LYCs with high and low muscle elasticity had the greatest differences in intestinal metabolites.

The first 20 DAMs in the intestines of the male LYCs were selected for Random Forest plot analysis, among which the 3.alpha.,4.beta.-galactotriose, Secoisolariciresinol, Tazarotenic acid, and Valeric acid were highly abundant in the IEHM group, while the rest of the intestinal metabolites showed low abundance. The DAMs in the IELM group showed the opposite results to those in the IEHM group (Figure 7a). 2’-O-Methyluridine, N6-methyladenine, Meperidine, Citrate, Cytidine 5’-monophosphate, and Man3 a, were highly abundant in the IEHF group, and the rest of the intestinal metabolites showed low abundance. The DAMs in the IELF group showed opposite results to those in the IEHF group (Figure 7b). Overall, the metabolic profiles of the intestinal contents differed significantly between the high and low muscle elasticity groups.

### 3.5. Metabolite Enrichment Analysis of the Intestinal Contents

To further explore the metabolic mechanism of muscle elasticity, Kyoto Encyclopedia of Genes and Genomes (KEGG) functional enrichment was performed, and the top 20 significantly enriched pathways were identified. A total of 46 metabolic pathways were enriched in the IEHM and IELM groups, among which the most significantly enriched metabolic pathways were: Metabolic pathways; Purine metabolism; Biosynthesis of cofactors; ABC transporters; Nicotinate and nicotinamide metabolism; Glycerophospholipid metabolism; Ascorbate and aldarate metabolism; Cysteine and methionine metabolism; Tyrosine metabolism; and Arginine biosynthesis (Figure 7c). Additionally, 44 metabolic pathways were enriched in the IEHF and IELF groups, among which the most predominant enriched metabolic pathways were Phenylalanine, tyrosine and tryptophan biosynthesis; Metabolic pathways; Citrate cycle (TCA cycle); Biosynthesis of cofactors; Glycosaminoglycan biosynthesis-heparan sulfate/heparin; Lysosome; Glycosaminoglycan biosynthesis-chondroitin sulfate/dermatan sulfate; and the FoxO signaling pathway (Figure 7d).

### 3.6. Gut Flora and Metabolite Association Analysis

The association analysis of the male LYC gut flora with metabolites (Figure 8a) showed that *Robertmurraya kyonggiensis*, *Solirubrobacter sp003344625*, *Agarivorans sp005405585*, *Photobacterium alginatilyticum*, and *Shewanella fidelis* correlated significantly or highly significantly positively (*p* < 0.05 or *p* < 0.01) with a large number of metabolites (more than 17 metabolites). *Vibrio harveyi*, *Methyloversatilis discipulorum A*, *Vibrio ponticus*, and *Phaeobacter italicus* correlated significantly or highly significantly negatively (*p* < 0.05 or *p* < 0.01) with a large number of metabolites (more than 12 metabolites).

As shown in Figure 8b, *Vibrio renipiscarius*, *Saccharomycodes ludwigii*, *Vibrio chagasii*, *Murine coronavirus*, and *Plasmodium berghei* were associated with a large number of metabolites (more than 17 metabolites) in the IEHF and IELF groups significant or highly significant positive correlations (*p* < 0.05 or *p* < 0.01). In addition, *Vibrio renipiscarius*, *Murine coronavirus*, *Saccharomycodes ludwigii*, *Vibrio splendidus F*, and *Plasmodium berghei* correlated significantly or highly significantly negatively with a large number of metabolites (more than 36 metabolites) (*p* < 0.05 or *p* < 0.01). Overall, the gut flora structure of LYCs was closely related to their nutrient metabolism.

## 4. Discussion

The LYC is the most cultured marine fish species in China, with the highest yield and the largest aquaculture area [38]. With improvements in living standards, endeavors to improve the taste of LYC have been gradually undertaken, and research on muscle elasticity of LYC has gradually become a focal point of current studies. The fish intestinal tract harbors numerous microorganisms, which exert a critical function in maintaining intestinal health, promoting body development, and regulating nutritional metabolism, the immune response, and muscle quality [31,39,40]. The intestinal flora might play an important role in improving the muscle quality of LYCs, thus, the present study aimed to build a relationship model between intestinal flora and muscle elasticity.

The interaction between the gut microbiota and muscle quality has been a focus of recent scientific research. Interestingly, it has been suggested that a stable and diverse gut microbiota can govern muscle development, growth, and function [41]. The muscles have the main function of contracting to facilitate the movement of the body, which involves energy expenditure; therefore, the existence of a gut–muscle communication pathway has been suggested [42]. An imbalance or a reduction in gut microbiota diversity could also lead to muscular dysplasia and dysfunction in fish, and bacteria-derived nutrients incorporated into muscles are related to the prevalence of bacterial taxa, in which their deficit is regarded as an early warning sign of microbiota disturbance [43]. Herein, we showed that there was a significant difference in the composition of the gut flora between the high and low muscle elasticity groups, indicating that the intestinal flora of LYC might regulate muscle elasticity. The intestinal flora has been shown to be closely related to differentiation of growth rates and muscle thickness in fish [22]. Evidence suggests that the gut microbiota composition and diversity can be a determinant of muscle metabolism and functionality [44]. Therefore, we inferred that gut microbes could affect the host development and promote muscle growth by regulating intestinal metabolism [45,46].

It has been demonstrated that fish muscle elasticity is higher, and its chewability is better, when Proteobacteria become the dominant intestinal bacteria, and the intestinal flora can improve muscle textural properties by influencing the regulation of fat metabolism, increasing friction between muscle bundles, and decreasing the triglyceride content of the muscle [47]. In this study, the dominant phylum in all groups was Proteobacteria, whose abundance was higher in the high muscle elasticity group. In the low muscle elasticity group, the abundance of Proteobacteria was low, which is consistent with the results of Qian et al. (2023), who showed that a significant change in Proteobacteria levels reduced the diameter of muscle fibers, increased the density of muscle fibers, and altered muscle texture (elasticity, hardness, cohesion, and chewing) in *hybrid snakeheads* [39]. Moreover, Proteobacteria have been shown to correlate positively with the contents of Thr, Lys, Pro, Asp, Gly, and Glu in muscle, and an association between Proteobacteria and higher amino acid deposition in muscle was observed in fish [48,49]. Therefore, it can be concluded that Proteobacteria play an important role in the process of muscle formation.

The gut flora directly or indirectly participates in body metabolism, regulating muscle growth and thus affecting muscle elasticity [50]. Research has confirmed that there is an inextricable link between gut metabolites and muscle quality [51], which is consistent with our results. Indeed, our findings showed that a close relationship was revealed between muscle elasticity and intestinal metabolism. Amino acid metabolism plays a crucial role in muscle growth and metabolic responses in teleosts [52,53], especially in muscle development, e.g., tryptophan is the limiting amino acid for protein synthesis and improves growth performance, meat quality, and protein deposition in fish [51,54]. Herein, a large number of enriched metabolic pathways related to muscle elasticity were identified for both male and female LYCs, with amino acid-based metabolic pathways being the most abundant, among which tryptophan biosynthesis has been shown to be an important intestinal metabolic pathway in LYCs. Thus, it was inferred that insufficient amino acid metabolism contributes to decreased protein synthesis, which in turn leads to decreased muscle elasticity.

In the present study, we demonstrated that Vibrio was correlated significantly or highly significantly negatively (*p* < 0.05 or *p* < 0.01) with a large number of metabolites in both females and males. Vibrio is a typical marine pathogen, which has been observed frequently to correlate negatively with muscle quality [55,56]. Vibriosis is one of the most prevalent bacterial diseases affecting LYCs [56]. Therefore, Vibrio might affect muscle quality of LYCs by causing metabolic disorders. The *Vibrio harveyi* was shown to correlate significantly and positively with a large number of metabolites in male LYCs. Studies have found that *Vibrio harveyi* causes muscle necrosis in fish [57]. Moreover, *Vibrio harveyi* infections have been identified frequently in LYCs. Thus, we hypothesized that *Vibrio harveyi* might affect muscle quality by causing metabolic disorders and muscle necrosis in LYCs [58].

## 5. Conclusions

We identified a large number of core gut flora and key metabolites associated with muscle elasticity by integrating macrogenomic and metabolomic techniques, thus providing insights into the molecular mechanisms explaining the differences in muscle quality in LYCs. The abundance of Streptophyta was significantly related to muscle elasticity. We further constructed an association model between gut flora and metabolites. The Vibrio was correlated significantly or highly significantly negatively with a large number of metabolites in both females and males. The result offers a new perspective on the potential mechanisms by which the LYC gut flora influences muscle quality via the regulation of intestinal metabolism. Additionally, our study not only provides new insights into the gut microbiota–metabolism–muscle axis, but also strong support for future research on muscle quality trait improvement in LYC.

## Figures and Tables

**Figure 1 animals-14-02690-f001:**
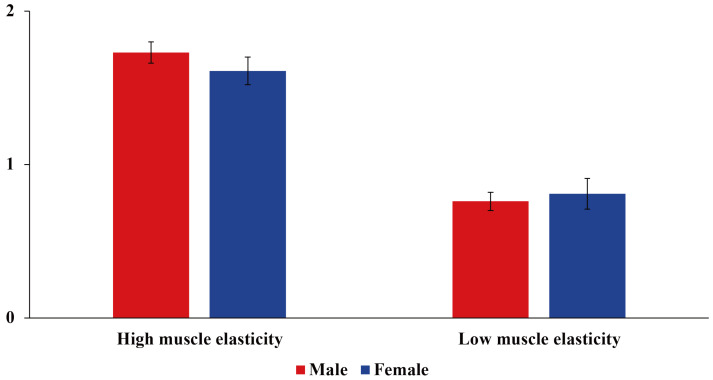
Muscle elastic characteristics of LYCs.

**Figure 2 animals-14-02690-f002:**
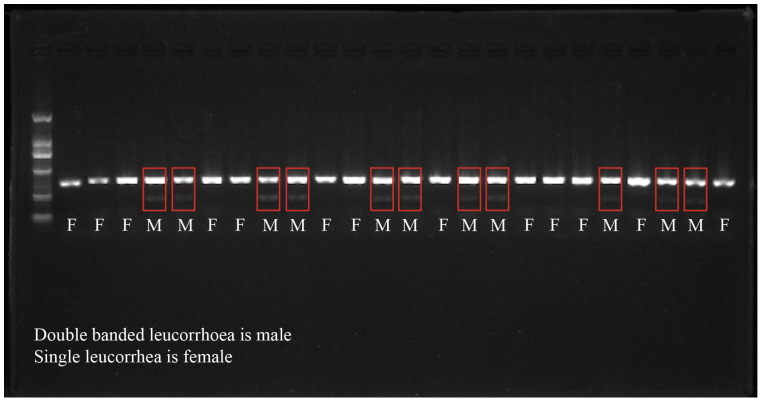
Agarose gel electrophoresis for sex determination in LYCs.

**Figure 3 animals-14-02690-f003:**
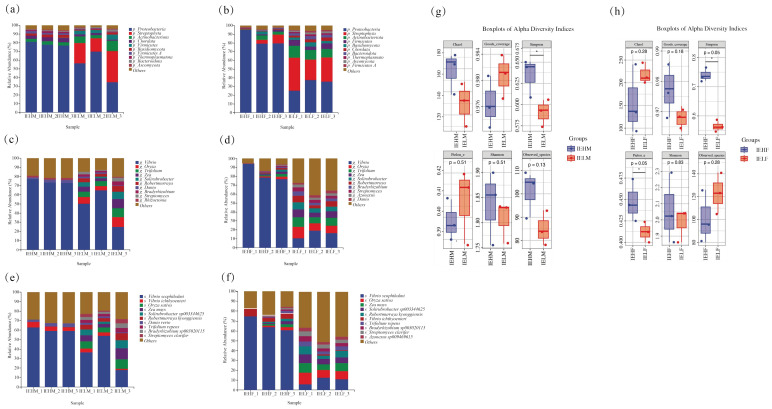
Structural analysis of the LYC intestinal flora. (**a**) Male LYC intestinal flora at the phylum level; (**b**) Female LYC intestinal flora at the phylum level; (**c**) Male LYC intestinal flora at the genus level; (**d**) Female LYC intestinal flora at the genus level; (**e**) Male LYC intestinal flora at the species level; (**f**) Female LYC intestinal flora at the species level; (**g**) Male LYC intestinal flora Alpha diversity analysis; (**h**) Female LYC intestinal flora Alpha diversity analysis. * *p* ≤ 0.05.

**Figure 4 animals-14-02690-f004:**
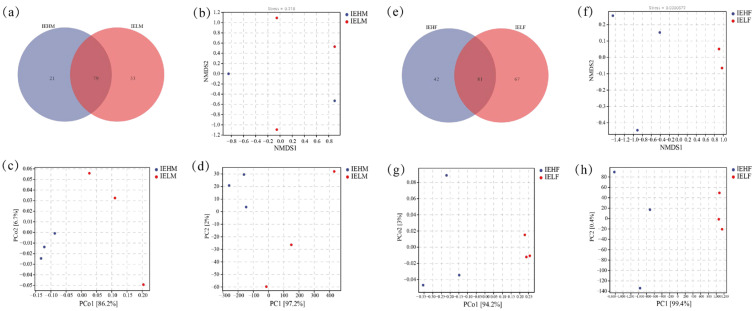
Analysis of the differences in the gut flora of LYC with different muscle elasticities: (**a**) Venn diagrams between the IEHM and IELM groups at the genus level; (**b**) Non-metric multidimensional scaling (NMDS) analysis of the intestinal flora in male LYCs; (**c**) Principal coordinate analysis (PCoA) of the intestinal flora in male LYCs; (**d**) Principal component analysis (PCA) of the intestinal flora in male LYCs; (**e**) Venn diagrams between the IEHF and IELF groups at the genus level.; (**f**) Non-metric multidimensional scaling (NMDS) analyses of the intestinal flora in female LYCs; (**g**) Principal coordinate analysis (PCoA) of the intestinal flora in female LYCs; (**h**) Principal component analysis (PCA) of the intestinal flora in female LYCs.

**Figure 5 animals-14-02690-f005:**
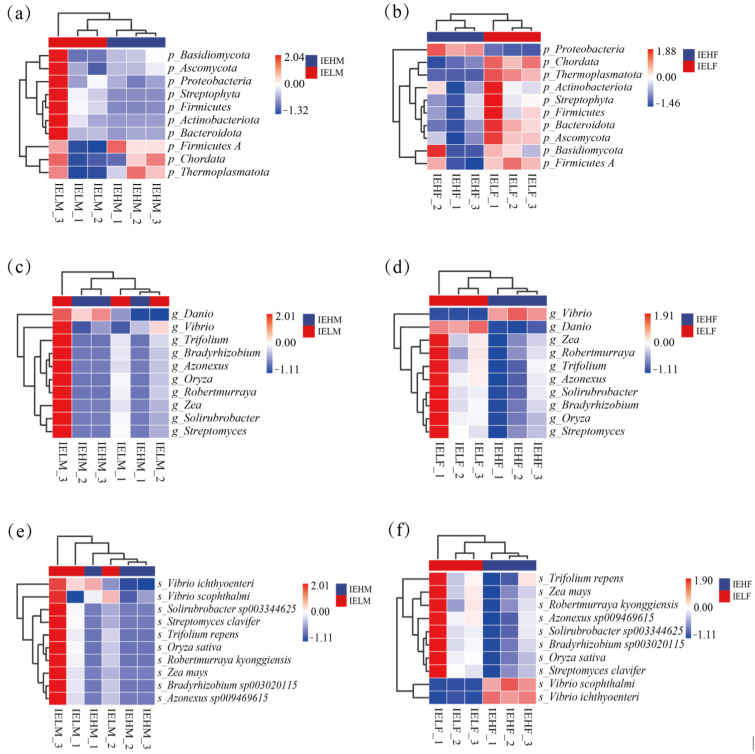
Heat map of the relative abundance of the gut flora. (**a**) Heat map of the relative abundance of the gut flora in males at the phylum level; (**b**) Heat map of the relative abundance of the gut flora in females at the phylum level; (**c**) Heat map of the relative abundance of the gut flora in males at the genus level; (**d**) Heat map of the relative abundance of the gut flora in females at the genus level; (**e**) Heat map of the relative abundance of the gut flora in males at the species level; (**f**) Heat map of the relative abundance of the gut flora in females at the species level; LYC, large yellow croaker; LYC, large yellow croaker; IEHM, intestinal contents of high muscle elasticity males; IELM, intestinal contents of low muscle elasticity males; IEHF, intestinal contents of high muscle elasticity females; IELF, intestinal contents of low muscle elasticity females.

**Figure 6 animals-14-02690-f006:**
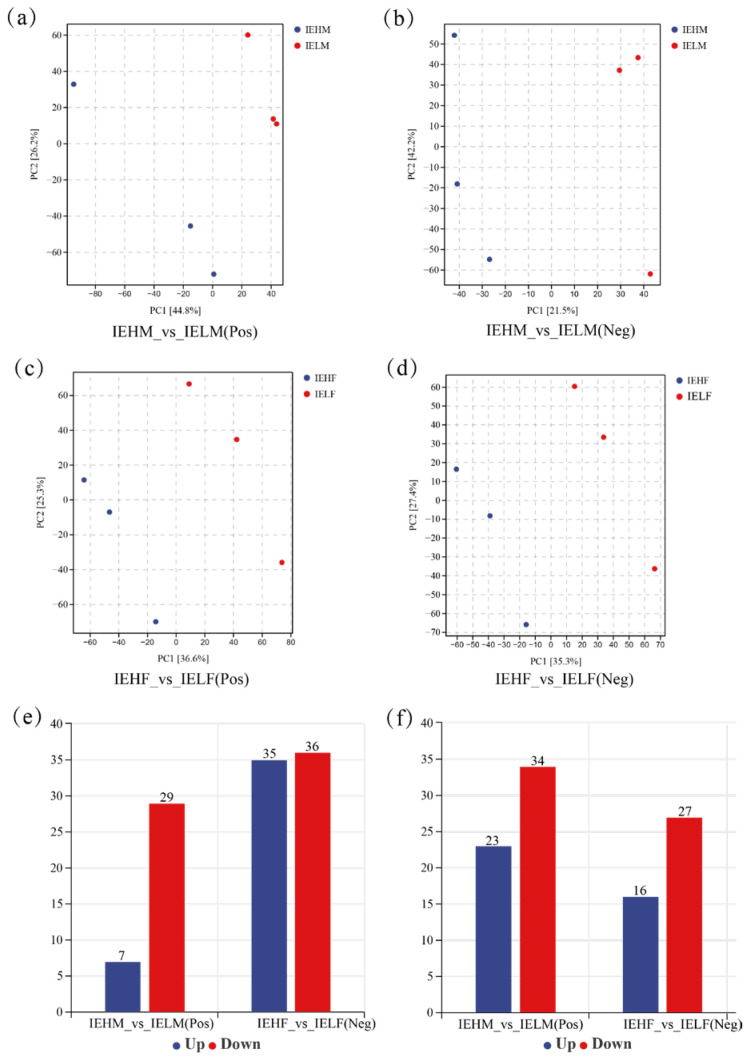
Analysis of LYC intestinal metabolites: (**a**) Intestinal metabolite analysis in male LYCs (Pos, positive ion mode); (**b**) Intestinal metabolite analysis in male LYCs (Neg, negative ion mode); (**c**) Intestinal metabolite analysis in female LYCs (positive ion mode); (**d**) Intestinal metabolite analysis in female LYCs (negative ion mode); (**e**) Differential metabolite analysis of LYC intestinal contents (positive ion mode); (**f**) Differential metabolite analysis of LYC intestinal contents (negative ion mode).

**Figure 7 animals-14-02690-f007:**
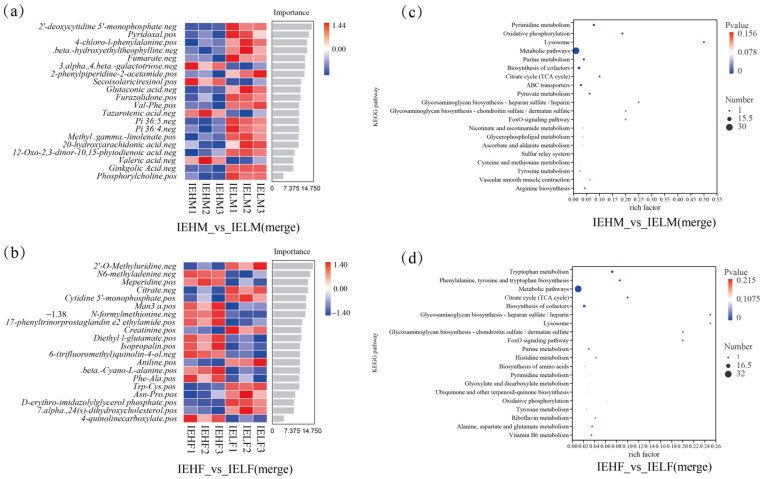
Analysis of LYC intestinal metabolites: (**a**) Random forest analysis of intestinal metabolites in male LYCs; (**b**) Random forest analysis of intestinal metabolites in female LYCs; (**c**) Differential metabolite enriched pathway analysis of intestinal contents in male LYCs; (**d**) Differential metabolite enriched pathway analysis of intestinal contents in female LYCs. LYC, large yellow croaker; IEHM, intestinal contents of high muscle elasticity males; IELM, intestinal contents of low muscle elasticity males; IEHF, intestinal contents of high muscle elasticity females; IELF, intestinal contents of low muscle elasticity females; PC2, principal component 2.

**Figure 8 animals-14-02690-f008:**
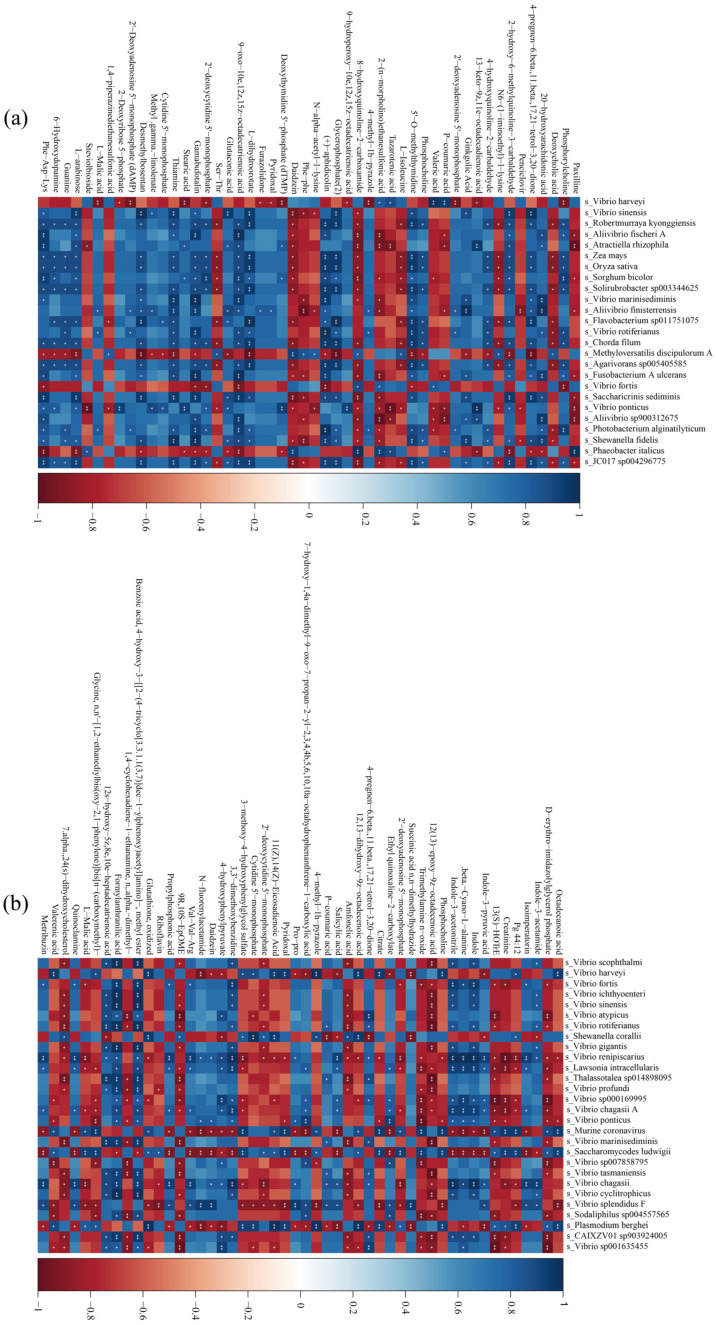
Association analysis of the LYC gut flora with differentially abundant metabolites. (**a**) male LYCs; (**b**) female LYCs. LYC, large yellow croak. The symbol * is the degree to which the metabolite is associated with a particular flora, where * is a significant association and ** is a very significant association. (Figure 8a,b).

## Data Availability

The data are available at: https://www.ebi.ac.uk/metabolights/editor/www.ebi.ac.uk/metabolights/MTBLS10457 (accessed on 10 January 2024).

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
