# Peer review of "Metagenomic and Metabolomic Analyses Reveal the Role of Gut Microbiome-Associated Metabolites in the Muscle Elasticity of the Large Yellow Croaker (Larimichthys crocea)"

_animals, 2024, doi:10.3390/ani14182690_

Round 1
Reviewer 1 Report
Comments and Suggestions for Authors
This is a very interesting work and a valuable contribution to an emerging topic. The paper presents background information on the relationship between the microbiome and muscle quality. It is suggested to improve the presentation of results to make it more reader-friendly. Additionally, it is recommended to strengthen the description of the results with the statistical analyses performed (significance). Below some suggestions:
Line 89:
2.1 Sample collection
It is suggested to include background information on the cultivation conditions such as pond type, density, water quality (temperature, oxygen concentration), cultivation system, diets used, and other factors that could be related to the microbiome.
Line 15: It is suggested to include the term LYCs in the summary.
Line 56: It is suggested to include a reference.
Line 59: It is suggested to include a reference.
Some references to consider:
Wen, C., Wang, Q., Gu, S., Jin, J., & Yang, N. (2024). Emerging perspectives in the gut–muscle axis: The gut microbiota and its metabolites as important modulators of meat quality. Microbial Biotechnology, 17(1), e14361.
Mancin, L., Wu, G. D., & Paoli, A. (2023). Gut microbiota–bile acid–skeletal muscle axis. Trends in microbiology, 31(3), 254-269.
Lien 92: It is suggested to define "suitable".
Line 94: It is suggested to describe how the elasticity of the fish was determined, prior to mentioning it in the methods section. This is a fundamental method for the interpretation of the study.
Figure 1: It is suggested to separate the elasticity and sex determination graphs included in Figure 1, under the description: Fig. 1. Structural analysis of the LYC intestinal flora. Include only the information that describes the figure.
It is suggested to increase the size of the graphs and legends.
Line 102: Please describe the controls used in PCR and confirmation methods for at least a percentage of the samples, through PCR product sequencing.
Line 124: It is suggested to include a reference.
Line 128: Please specify and objectively describe: “appropriate amount”.
Line 160: It is suggested to separate the graphs in relation to the presentation of the results, as the figure appears too crowded with many graphs.
Line 265: It is suggested to separate the graphs in relation to the presentation of the results, as the figure appears too crowded with many graphs.
Best regards,
Author Response
Reply to reviewers
Manuscript Number:
Reviewer: 1
1.Line 89:
2.1 Sample collection
It is suggested to include background information on the cultivation conditions such as pond type, density, water quality (temperature, oxygen concentration), cultivation system, diets used, and other factors that could be related to the microbiome.
Reply: We have provided additional explanations, as followed:
All farmed LYCs obained from the same nursery were cultured in the same sea area, using the same feed.
2.Line 15: It is suggested to include the term LYCs in the summary.
Reply: We have added the the term “Large yellow croaker (LYC, Larimichthys crocea)” in the summary.
3.Line 56: It is suggested to include a reference.
Reply: A reference has been added as suggested.
4.Line 59: It is suggested to include a reference.
Reply: A reference has been added as suggested.
5.Lien 92: It is suggested to define "suitable".
Reply: We have modified it.
6.Line 94: It is suggested to describe how the elasticity of the fish was determined, prior to mentioning it in the methods section. This is a fundamental method for the interpretation of the study.
Reply: Determination of texture characteristics (muscle elasticity) : The back muscle of large yellow croaker was sampled and cut into 2 cm x 2 cm x 1 cm squares.The the elasticity of muscle were measured by TMS-PRO texture analyzer. Extrusion test parameters are as follows: probe pause time 1s, range of force sensing element 1000N, probe rebound height 25mm, shape variable 50%, Measuring speed 60mm/min, starting force 1.5N.
7.Figure 1: It is suggested to separate the elasticity and sex determination graphs included in Figure 1, under the description: Fig. 1. Structural analysis of the LYC intestinal flora. Include only the information that describes the figure.
Reply: We haved separated the elasticity and sex determination graphs according to your suggestions.
8.It is suggested to increase the size of the graphs and legends.
Reply: We haved increased the size of the graphs and legends according to your suggestions.
9.Line 102: Please describe the controls used in PCR and confirmation methods for at least a percentage of the samples, through PCR product sequencing.
Reply: We have made a detailed supplementary explanation, as followed:
The amplification products were identified using 1.5% agarose gel electrophoresis (specific conditions: voltage 220 volts, time 25 minutes, buffer type 1x TAE). The specific conditions of PCR amplification were as follwed: predenaturation 95℃, denaturation 94℃, annealing temperature 64℃, extension 72℃, cycle 29 times, re-extension 72℃, preservation 4℃。
10.Line 124: It is suggested to include a reference.
Reply: A reference has been added according to your suggestions.
11.Line 128: Please specify and objectively describe: “appropriate amount”.
Reply: We have modified it.
12.Line 160: It is suggested to separate the graphs in relation to the presentation of the results, as the figure appears too crowded with many graphs.
Reply: We have made adjustments to the graphs.
13.Line 265: It is suggested to separate the graphs in relation to the presentation of the results, as the figure appears too crowded with many graphs.
Reply: We have made adjustments to the graphs.

Reviewer 2 Report
Comments and Suggestions for Authors
By integrating metamogenics and metabolomics, the authors revealed the key role of gut flora and metabolites in muscle elasticity of Larimichthys crocea, providing a new perspective on the molecular mechanism of muscle mass differences in L. crocea, and strengthening the understanding of gut microbiome - metabolism - muscle axis, supporting future related research. However, some aspects are relatively simple, and it is suggested to strengthen the research in some aspects.
Here, I would like to make some suggestions to further improve this article:
Simple Summary
The brief overview, abstract, introduction, and body are separate parts of the article, and when the species name appears in each separate part, the full Latin name is used for the first time, followed by an abbreviated form. The brief overview is a concise summary of the whole article. Please explain the meaning of "LYC" first in this section, and then use abbreviations.
Abstract
Page 1, lines 21-24. It is suggested that the purpose of this study should be further explained in this section, rather than simply describing the relationship between intestinal flora and muscle elasticity with these research backgrounds.
Introduction
1. Page 2, line 46. When the species name first appears, complete the name and permissions of the species, including the first mention of the species (e.g. Dicentrachus labrax Linaeus 1758).
2. Page 2, lines 49-55. It is suggested that after describing the economic value, distribution and farming mode of LYC, a more direct transition to the current problems of farming, such as germplasm degradation, disease and quality decline; The paper emphasizes the impact of these problems on the sustainable development of LYC industry, and then leads to the necessity and urgency of improving LYC muscle mass research, which makes the logic of the paper more compact.
3. Page 2, line 62. What is the reference basis for the description? Please cite relevant literature to support their views.
4. Page 2, lines 75-76. It is suggested to provide support and supplement for the view that "there are few studies on the regulation of muscle elasticity by intestinal flora", and to support the view by citing research literature in related fields to increase the persuasive strength of the argument.
5. Page 2, lines 82-86. The specific purpose of this study was clearly stated, that is, using non-targeted metabolomics and macrogenomics technologies, to analyze how the LYC intestinal flora and nutrient metabolism work together to regulate muscle elasticity. It is suggested that this paragraph further elaborate the purpose and significance of the study.
Materials and methods
1. Page 3, lines 89-91. Please further describe the source of the sample, provide the geographical coordinates of the sampling points, and give me a map of the above area so that I can more accurately determine the location of the sample.
2. Page 3, lines 105-107. It is recommended to provide the specific conditions (such as voltage, electrophoresis time, buffer type, etc.) of 1% agarose gel electrophoresis and the specific conditions of PCR amplification, including annealing temperature, elongation time, cycle number, etc., which are crucial for the reproduction of experimental results.
3. Page 3, lines 113-114 and 126. Please confirm whether the company name is the correct company name and ensure the accuracy of the company name to avoid confusion.
4. Page 3, lines 122-124, 139-140. Should this part of the content be placed in the following "Data analysis" chapter, for a detailed description of how to process, analyze and interpret the collected data?
5. Page 4, line 157. "LYC, large yellow croaker", repeated.
Result
1. Page 5, line 190. Please check whether the Latin scientific names of intestinal flora such as "Vibrio" in the full text need to be italicized, and unification of the full text, because there are many inconsistencies.
2. Page 6, line 252. "s_", please unify this part with the following text.
Discussion
1. Page 9, lines 351-359. Which is more of the content of the introduction, only explains that this study aims to explore the microbial regulatory pathway for improving the muscle mass of this fish species by establishing a relationship model between intestinal flora and LYC meat elasticity, but this is not a complete way to discuss the results. In order to better understand and evaluate the results of the study, it is suggested that a comprehensive analysis combined with other studies is very necessary, so please consider whether this paragraph is necessary and re-write it.
2. Page 10, lines 369-371. When referring to "Results indicate", please further elaborate on the specific differences in intestinal flora composition between the high and low myoelastic groups, including which bacteria are more abundant or scarce, and how these differences are related to muscle elasticity, which will strengthen the findings.
3. Page 10, lines 374-376. In view of the author's inference, although based on existing research, but please express more carefully. It is recommended to add supporting evidence for this inference, such as citing more specific experimental data or theoretical models.
4. Page 10, lines 387-388. Simplify the sentence structure, avoid lengthy and complex expression. For example, "Proteobacteria has been shown to be associated with Thr, Lys, Pro, Asp, Gly, and glutamate" could be simplified to "Proteobacteria is associated with the metabolism of multiple amino acids, including Thr, Lys, Pro, etc."
5. Page 10, lines 394-396. "Study confirms that there is an inextricably linked link between intestinal metabolites and muscle mass" partially overlaps with the previous article, "Our findings suggest significant differences in intestinal metabolic characteristics between high and low muscle elasticity LYCs," and the language should be simplified or reorganized.
6. Page 10, lines 400-403. The authors mention the similarities in metabolic pathways between male and female LYCs, but in light of the earlier use of SNP tests specifically to distinguish the sex of LYCs. Therefore, further discussion on whether there are sex-specific differences is recommended. For example, whether certain amino acid metabolic pathways or intestinal flora composition differ significantly between males and females, and how these differences affect muscle elasticity.
7. Page 10, lines 406-411. Although the author has proposed the association of Vibrio with LYC metabolic disorders and muscle necrosis, it should be further clarified whether this association directly leads to the decline in muscle mass. Please cite more specific literature support to reinforce this relationship.
8. Page 10, lines 412-416. It is recommended to cite more literature to support the author's point of view, especially the literature on the relationship between Vibrio and muscle mass, to further enhance the credibility of the argument.
Conclusions
The conclusion of an article should be a further summary and distillation of the research results, and it should clearly and deeply explain the significance and impact of the research findings. Based on your previous content, I suggest rewriting the conclusion. For example, specifying which specific gut microbiota and metabolites have a significant effect on muscle elasticity further provides more specific information about the association model. In addition, based on the current research results, future research questions or directions worth exploring are proposed to promote the further development of related fields.
Other
1. Page 11, line 427. The following statements should be used..... Please check??????
2. Please attach a statement to your article stating what ethical approvals or other relevant permissions were obtained for the research. If applicable, the statement should include the name of the relevant ethics committee or body that provided the approval or permit/license, and a possible reference number. If the study is granted an exemption from requiring ethical approval, it should be stated along with the name of the ethics committee providing the exemption and/or the reason for the exemption.
3. Please refer to the relevant journal guide to determine whether to add acknowledgements and other content.
4. In view of some grammatical problems in the manuscript, it is necessary to polish it to improve the accuracy and academicity of its language expression.
Reference
Please update the literature with the latest research. Also, refer to the guidelines for writing bibliographic entries and textual citations. For name items that require special characters, pay attention to writing symbols to avoid miswriting the author.
Comments on the Quality of English LanguageExtensive editing of English language required.
Author Response
Reply to reviewers
Manuscript Number:
Reviewer: 1
Simple Summary
The brief overview, abstract, introduction, and body are separate parts of the article, and when the species name appears in each separate part, the full Latin name is used for the first time, followed by an abbreviated form. The brief overview is a concise summary of the whole article. Please explain the meaning of "LYC" first in this section, and then use abbreviations.
Reply: It has been modified as suggested.
Abstract
Page 1, lines 21-24. It is suggested that the purpose of this study should be further explained in this section, rather than simply describing the relationship between intestinal flora and muscle elasticity with these research backgrounds.
Reply: It has been modified as suggested.
Introduction
- Page 2, line 46. When the species name first appears, complete the name and permissions of the species, including the first mention of the species (e.g. Dicentrachus labrax Linaeus 1758).
Reply:It has been modified as suggested
- Page 2, lines 49-55. It is suggested that after describing the economic value, distribution and farming mode of LYC, a more direct transition to the current problems of farming, such as germplasm degradation, disease and quality decline; The paper emphasizes the impact of these problems on the sustainable development of LYC industry, and then leads to the necessity and urgency of improving LYC muscle mass research, which makes the logic of the paper more compact.
Reply: The structure of the article has been improved to make our paper more logical according to the suggestions.
- Page 2, line 62. What is the reference basis for the description? Please cite relevant literature to support their views.
Reply: A reference has been added as recommended.
- Page 2, lines 75-76. It is suggested to provide support and supplement for the view that "there are few studies on the regulation of muscle elasticity by intestinal flora", and to support the view by citing research literature in related fields to increase the persuasive strength of the argument.
Reply: A reference has been added according to your suggestions.
- Page 2, lines 82-86. The specific purpose of this study was clearly stated, that is, using non-targeted metabolomics and macrogenomics technologies, to analyze how the LYC intestinal flora and nutrient metabolism work together to regulate muscle elasticity. It is suggested that this paragraph further elaborate the purpose and significance of the study.
Reply: We have reorganized the purpose and meaning of our article according to your suggestions.
Materials and methods
- Page 3, lines 89-91. Please further describe the source of the sample, provide the geographical coordinates of the sampling points, and give me a map of the above area so that I can more accurately determine the location of the sample.
Reply: Fujian Fuding Aquaculture Experimental Center of East China Sea Fisheries Research Institute is located in Jiayang Township, Fuding City, Fujian Province. It covers a total area of 3.45 hectares, including 2.06 hectares of land and 1.39 hectares of sea. In 2009, it was supported by the basic project of the Ministry of Agriculture and put into use in 2014. Fuding Test base has a comprehensive experiment building, 2 factory breeding workshops with a total of 1500m2, 2 seed cultivation workshops with a total of 1500m2, outdoor cement pool, outdoor soil pool, physiological and ecological experiment workshop and bioberbium material culture workshop, equipped with conventional scientific research equipment.
Add: No. 24, Xiajing, 971 County Road, Fuding City, Ningde City, Fujian Province
Geographical coordinates: 120.40431,27.222208
- Page 3, lines 105-107. It is recommended to provide the specific conditions (such as voltage, electrophoresis time, buffer type, etc.) of 1% agarose gel electrophoresis and the specific conditions of PCR amplification, including annealing temperature, elongation time, cycle number, etc., which are crucial for the reproduction of experimental results.
Reply: We have provided additional explanations according to your suggestions.
- Page 3, lines 113-114 and 126. Please confirm whether the company name is the correct company name and ensure the accuracy of the company name to avoid confusion.
Reply: We have corrected it.
- Page 3, lines 122-124, 139-140. Should this part of the content be placed in the following "Data analysis" chapter, for a detailed description of how to process, analyze and interpret the collected data?
Reply: We thought it was more appropriate to put this part of the content here, so that readers could easily understand the experimental methods and techniques of this study.
- Page 4, line 157. "LYC, large yellow croaker", repeated.
Reply: It has been corrected according to your suggestions.
Result
- Page 5, line 190. Please check whether the Latin scientific names of intestinal flora such as "Vibrio" in the full text need to be italicized, and unification of the full text, because there are many inconsistencies.
Reply: Yes, it's in italics. We have revised the full text.
- Page 6, line 252. "s_", please unify this part with the following text.
Reply: It has been modified according to your suggestions.
Discussion
- Page 9, lines 351-359. Which is more of the content of the introduction, only explains that this study aims to explore the microbial regulatory pathway for improving the muscle mass of this fish species by establishing a relationship model between intestinal flora and LYC meat elasticity, but this is not a complete way to discuss the results. In order to better understand and evaluate the results of the study, it is suggested that a comprehensive analysis combined with other studies is very necessary, so please consider whether this paragraph is necessary and re-write it.
Reply: This paragraph does not focus on the results of this research, but gives a brief introduction to the current research background. We believe that this paragraph is necessary here.
- Page 10, lines 369-371. When referring to "Results indicate", please further elaborate on the specific differences in intestinal flora composition between the high and low myoelastic groups, including which bacteria are more abundant or scarce, and how these differences are related to muscle elasticity, which will strengthen the findings.
Reply: The specific differences in intestinal flora and the correlation with muscle elasticity have been described in detail later in this article, so there is no redundancy here.
- Page 10, lines 374-376. In view of the author's inference, although based on existing research, but please express more carefully. It is recommended to add supporting evidence for this inference, such as citing more specific experimental data or theoretical models.
Reply: We have revised this sentence and added the corresponding references.
- Page 10, lines 387-388. Simplify the sentence structure, avoid lengthy and complex expression. For example, "Proteobacteria has been shown to be associated with Thr, Lys, Pro, Asp, Gly, and glutamate" could be simplified to "Proteobacteria is associated with the metabolism of multiple amino acids, including Thr, Lys, Pro, etc."
Reply: Your suggestion is very good, and we have revised it according to your suggestions.
- Page 10, lines 394-396. "Study confirms that there is an inextricably linked link between intestinal metabolites and muscle mass" partially overlaps with the previous article, "Our findings suggest significant differences in intestinal metabolic characteristics between high and low muscle elasticity LYCs," and the language should be simplified or reorganized.
Reply: We have simplified this part of content.
- Page 10, lines 400-403. The authors mention the similarities in metabolic pathways between male and female LYCs, but in light of the earlier use of SNP tests specifically to distinguish the sex of LYCs. Therefore, further discussion on whether there are sex-specific differences is recommended. For example, whether certain amino acid metabolic pathways or intestinal flora composition differ significantly between males and females, and how these differences affect muscle elasticity.
Reply: The genetic background information of females and males is very different, so we did not study the differences between them in order to eliminate genetic information interference.
- Page 10, lines 406-411. Although the author has proposed the association of Vibrio with LYC metabolic disorders and muscle necrosis, it should be further clarified whether this association directly leads to the decline in muscle mass. Please cite more specific literature support to reinforce this relationship.
Reply: A reference has been added as recommended.
- Page 10, lines 412-416. It is recommended to cite more literature to support the author's point of view, especially the literature on the relationship between Vibrio and muscle mass, to further enhance the credibility of the argument.
Reply: A reference has been added as recommended.
Conclusions
The conclusion of an article should be a further summary and distillation of the research results, and it should clearly and deeply explain the significance and impact of the research findings. Based on your previous content, I suggest rewriting the conclusion. For example, specifying which specific gut microbiota and metabolites have a significant effect on muscle elasticity further provides more specific information about the association model. In addition, based on the current research results, future research questions or directions worth exploring are proposed to promote the further development of related fields.
Reply: We provided a brief supplementary explanation according to your suggestions.
Other
- Page 11, line 427. The following statements should be used..... Please check??????
Reply: Thank you very much for your suggestion, we have removed this sentence.
- Please attach a statement to your article stating what ethical approvals or other relevant permissions were obtained for the research. If applicable, the statement should include the name of the relevant ethics committee or body that provided the approval or permit/license, and a possible reference number. If the study is granted an exemption from requiring ethical approval, it should be stated along with the name of the ethics committee providing the exemption and/or the reason for the exemption.
Reply: We have already added this.
- Please refer to the relevant journal guide to determine whether to add acknowledgements and other content.
Reply: We have added the acknowledgements.
- In view of some grammatical problems in the manuscript, it is necessary to polish it to improve the accuracy and academicity of its language expression.
Reply: We have checked and improved our article.
Reference
Please update the literature with the latest research. Also, refer to the guidelines for writing bibliographic entries and textual citations. For name items that require special characters, pay attention to writing symbols to avoid miswriting the author.
Reply: We have updated our references and checked the format.

Round 2
Reviewer 2 Report
Comments and Suggestions for Authors
After carefully reviewing your revised submission on the effect of gut microbiota and metabolites
on the muscle elasticity of Larimichthys crocea, I feel that your study has important innovation
and potential in integrating metamogenics and metabolomics to reveal the relationship between
gut microbiota and muscle mass.
However, despite the logical clarity of your first revision, I unfortunately found that several key
review comments were not adequately responded to or elaborated upon, which affected the overall
quality and persuasiveness of the manuscript. In view of the above reasons, I believe that the
current manuscript is still insufficient in the treatment of key issues, and it is difficult to meet the
standards of publication. I therefore suggest that careful revision is still needed.
Simple Summary
1. Page 1, line 13, please check the details of the first line of this section.
Introduction
1. Page 2, line 46, verify whether the opinion has been modified, please carefully check this part
of the content, and make necessary modifications as needed. In addition, please submit a relevant
Cover letter when you resubmit your manuscript to better track and document the revision
process.
2. Page 2, lines 49-55, I find that this part needs further verification/revision. Please carefully
check this part of the content, confirm its accuracy, and make necessary changes as necessary. If
the author considers that the opinion does not need to be amended, please also state clearly why.
Please submit a Cover letter when you resubmit your manuscript!!
Materials and methods
1. Page 3, lines 89-91, please further describe the source of the sample, i.e. the geographic
coordinates of the same sea area.
2. Page 3, lines 132-134 and 149-150, the author is requested to clearly provide reasons why the
opinion does not need to be amended, and to submit a relevant Cover letter when resubmitting the
manuscript.
Discussion
1. Page 9, lines 351-359, this part is more about the content of the introduction, please consider
whether this paragraph is necessary. In response to this section, the author is requested to clearly
provide detailed reasons why the opinion does not need to be amended.
2. Page 10, lines 369-371, the author is requested to clearly provide the detailed reasons why this
opinion does not need to be amended.
3. Page 10, lines 387-388, the author is requested to clearly provide detailed reasons why this
opinion does not need to be amended.
4. Page 10, lines 400-403, the author clearly provide the detailed reasons why this opinion does
not need to be amended.
Other
Please provide further proof of the polishing of this manuscript.
I encourage you to make further revisions and additions to the manuscript according to the
comments of the reviewers. In particular, focus on the key points mentioned above and respond to
the reviewer's concerns by providing new in-depth analysis or detailed explanations whenever
possible. I believe that with your efforts, this research will eventually be published and have an
important academic impact.

Comments on the Quality of English LanguageModerate editing of the English language and further proof of appropriate embellishments is required.
Author Response
Cover letter
Manuscript Number:animals-3136540-peer-review-v2
Reviewer:
After carefully reviewing your revised submission on the effect of gut microbiota and metabolites
on the muscle elasticity of Larimichthys crocea, I feel that your study has important innovation
and potential in integrating metamogenics and metabolomics to reveal the relationship between
gut microbiota and muscle mass.
However, despite the logical clarity of your first revision, I unfortunately found that several key
review comments were not adequately responded to or elaborated upon, which affected the overall
quality and persuasiveness of the manuscript. In view of the above reasons, I believe that the
current manuscript is still insufficient in the treatment of key issues, and it is difficult to meet the
standards of publication. I therefore suggest that careful revision is still needed.
Simple Summary
- Page 1, line 13, please check the details of the first line of this section.
Introduction
Reply: We have checked the detials of this section, and a format error was corrected.
- Page 2, line 46, verify whether the opinion has been modified, please carefully check this part
of the content, and make necessary modifications as needed. In addition, please submit a relevant
Cover letter when you resubmit your manuscript to better track and document the revision
process.
Reply: It has been modified as suggested.
- Page 2, lines 49-55, I find that this part needs further verification/revision. Please carefully
check this part of the content, confirm its accuracy, and make necessary changes as necessary. If
the author considers that the opinion does not need to be amended, please also state clearly why.
Please submit a Cover letter when you resubmit your manuscript!!
Materials and methods
Reply: We made appropriate adjustments. In this part, we have introduced in detail the status quo, challenges, solutions, and research status of large yellow croaker, and we believe that this narrative is reasonable and appropriate. It is hoped that readers will have a more detailed understanding of the various aspects of the research object.
- Page 3, lines 89-91, please further describe the source of the sample, i.e. the geographic
coordinates of the same sea area.
Reply: Thanks for your suggestion, I have added the geographical coordinates.
- Page 3, lines 132-134 and 149-150, the author is requested to clearly provide reasons why the
opinion does not need to be amended, and to submit a relevant Cover letter when resubmitting the
manuscript.
Reply: The experimental steps are carried out in strict accordance with the procedures of the testing company, so we have added experimental steps so that more readers can repeat our experiment. And, these experiments are common, so we did not go into great detail.
Discussion
- Page 9, lines 351-359, this part is more about the content of the introduction, please consider
whether this paragraph is necessary. In response to this section, the author is requested to clearly
provide detailed reasons why the opinion does not need to be amended.
Reply:This paragraph does not focus on the results of this research, but gives a brief introduction to the current research background. We believe that this paragraph is necessary here.
- Page 10, lines 369-371, the author is requested to clearly provide the detailed reasons why this
opinion does not need to be amended.
Reply:Your insights and suggestions are excellent.The specific differences in intestinal flora and the correlation with muscle elasticity have been described in detail later in this article, so there is no redundancy here.We put the discussion you said in the later part to make the whole of the article more intact, so that readers can read the article more clearly.
- Page 10, lines 387-388, the author is requested to clearly provide detailed reasons why this
opinion does not need to be amended.
Reply:Your suggestion is very good, and we have revised it according to your suggestions.
Simplification is very necessary and has been modified.
- Page 10, lines 400-403, the author clearly provide the detailed reasons why this opinion does
not need to be amended.Other Please provide further proof of the polishing of this manuscript.
I encourage you to make further revisions and additions to the manuscript according to the
comments of the reviewers. In particular, focus on the key points mentioned above and respond to
the reviewer's concerns by providing new in-depth analysis or detailed explanations whenever
possible. I believe that with your efforts, this research will eventually be published and have an
important academic impact.
Reply:The genetic background information of females and males is very different, so we did not study the differences between them in order to eliminate genetic information interference.
Other
Please provide further proof of the polishing of this manuscript.
Reply:Before the submission, our article was edited and polished by one English language editing company called “ELIXIGEN CO. LTD”. We further checked the article carefully to make sure there were no grammatical and lexical errors. If you think the technical and grammatical revisions have not been made successfully, we will revise this paper again carefully.